# Food Choice Motives and COVID-19 in Belgium

**DOI:** 10.3390/foods11060842

**Published:** 2022-03-15

**Authors:** Elly Mertens, Diana Sagastume, Tamara Sorić, Ivona Brodić, Ivan Dolanc, Antonija Jonjić, Eva Anđela Delale, Mladen Mavar, Saša Missoni, Miran Čoklo, José L. Peñalvo

**Affiliations:** 1Unit of Non-Communicable Diseases, Institute of Tropical Medicine, Nationalestraat 155, 2000 Antwerp, Belgium; dsagastume@itg.be (D.S.); jpenalvo@itg.be (J.L.P.); 2Psychiatric Hospital Ugljan, Otočkih dragovoljaca 42, 23275 Ugljan, Croatia; novoselic.tamara@gmail.com (T.S.); ivona.nutricionist@gmail.com (M.M.); 3Nutrition ID Ltd., Vranovina 30, 10000 Zagreb, Croatia; ravnatelj@pbu.hr; 4Centre for Applied Bioanthropology, Institute for Anthropological Research, Ljudevita Gaja 32, 10000 Zagreb, Croatia; ivan.dolanc@inantro.hr (I.D.); antonija.jonjic@inantro.hr (A.J.); miran.coklo@inantro.hr (M.Č.); 5Institute for Anthropological Research Ljudevita Gaja 32, 10000 Zagreb, Croatia; eva.andela.delale@inantro.hr (E.A.D.); sasa.missoni@inantro.hr (S.M.)

**Keywords:** COVID-19 pandemic, Belgium, food choice motives, Food Choice Questionnaire, sensory appeal, convenience, health

## Abstract

To investigate the main motives driving dietary intake changes potentially introduced by preventive measures to address the pandemic, an online survey, using a 36-item Food Choice Questionnaire applied for the period before (the year 2019) and during (2020–2021) the pandemic, was distributed between July and October 2021 among adult residents from Belgium. A total of 427 eligible respondents, the majority Dutch-speaking, were included for analyses. The importance of nine motives for food choices, including health, mood, convenience, sensory appeal, natural content, price, weight control, familiarity and ethical concerns, was assessed by scoring from 1 to 5, and comparing mean scores from the during period with the before period. Sensory appeal was the most important food choice motive before (mean score of 4.02 ± 0.51) and during (3.98 ± 0.48) the pandemic. Convenience and health also ranked among the main motives, with health observed to become more important during the pandemic (3.69 ± 0.60 during vs. 3.64 ± 0.59 before). Additionally, mean scores of mood (3.41 ± 0.71 vs. 3.32 ± 0.58), natural content (3.35 ± 0.84 vs. 3.26 ± 0.85) and weight control (3.33 ± 0.79 vs. 3.25 ± 0.76) were significantly higher during as compared to before. The extent of change in the level of importance for natural content was smaller with increasing age, and for health larger for urban areas, but for other motives there were no significant differences across population subgroups. Changes in the level of importance were observed in both directions, while a moderate share of respondents declared no change, suggesting some persistence of food choice motives. Further activities within public health monitoring should be considered to fully understand the COVID-19 implications on food choice motives together with people’s food behaviors and consumption.

## 1. Introduction

Since its outbreak in December 2019, the coronavirus disease (COVID-19) has been a major cause of morbidity and mortality worldwide. Public health measures of confinement have been introduced to limit the spread of the virus and protect the public from exposure to the infection. Mandatory isolation and remote work severely disrupt the individuals’ daily routines [1], likely resulting in decreasing levels of physical activity [2] and increasing sedentary behavior [3,4], as well as increasing frequency of meals and snacking [5,6]. Yet, the opposite has been also observed, for instance, a decrease in the frequency of ordered food and fast-food consumption [6], while an increase has been seen in the frequency of home cooking [6,7,8,9,10,11,12], which is often, but not necessarily, associated with increased consumption of fresh produce, mostly fruit and vegetables, which are examples of frequently observed dietary changes in the favorable direction. This heterogeneity in results has been recognized in a recent systematic review that found, for most European countries, an overall increase in healthful food practices of more home-made foods and less take-away/delivered food counterbalanced by an increase in unhealthful food practices of increased meal frequency and snacking as well as a tendency to consume more unhealthy foods and alcoholic beverages [13].

In addition, alarming increments in body weight have been observed during the lock-down in a significant proportion of the population [14], probably due to abruptly reduced physical activity and overeating arising from strict confinement measures during the COVID-19 pandemic, and potentially leading to higher incidence of overweight, obesity and relevant comorbidities [15]. Early findings from Belgium also showed that during the first six weeks of the confinement, only half of the adult population reported to have kept their body weight, while around 15% reported to have lost weight, and almost one-third gained weight with increased consumption of sweet or salty snacks and less physical activities as the most important correlates [16]. Strategies supporting physical activity and healthy diet, including meal planning, should therefore be implemented to encourage people to maintain or to return to a healthy lifestyle routine [17,18]. Given this controversy seen in the dietary behaviors and associated body weight changes, induced during the pandemic, there is a need to further understand what are the motives that drive specific food choices and how these might have changed as a result of the restrictive measures introduced by the COVID-19 pandemic.

To date, only a limited number of studies have addressed food choice motives showing that health was consistently nominated as a more important reason to select a specific food item during the pandemic as compared to before, along with weight control [19,20,21,22] which also relates to health. However, contradictory findings have been reported for the other highly ranked food choice motives: mood and sensory appeal becoming less important in Polish adolescents [19] and Croatian men [22], but more in French adults [20] and British adults (except sensory appeal that remained unchanged) [21]. Convenience, familiarity and price became more important in Croatian adults [22], but less in French [20] and British adults (except price more) [21], and no changes for Polish adolescents [19]. This illustrates that, as for food consumption, motives for food choices also depend on the population investigated and changes induced by the pandemic often occurring in both directions. More granular data on the individual characteristics that may influence the change might shed some light on these variable findings.

The aim of the present study is, therefore, to evaluate the extent of changes in food choice motives during the COVID-19 pandemic (RQ1) and to explore how these are related to individual characteristics in a study population of adults from Belgium (RQ2). The comprehensive understanding of who may be at risk for a change in food choice motives in the unfavorable direction would certainly support tailored health promotion programs aiming at tackling diet-related non-communicable diseases.

## 2. Materials and Methods

### 2.1. Study Design and Study Population

This cross-sectional study was conducted online through an electronic survey using voluntary sampling. An online survey was launched 2 July 2021 (four months after the onset of the third wave in Belgium) and was active until 31 October 2021, via different channels of social media including the Institute of Tropical Medicine (ITM) webpage, Twitter, Instagram, Facebook, and LinkedIn accounts, and by emailing academic and research institutes networks asking to further distribute to their contacts. Eligible participants were adults over 18 years of age, residents of Belgium and with access to a computer, tablet or phone with an internet connection. All participants were informed on the purpose of the study, assured of strict confidentiality and requested for their consent before participation. The hypothesis of this study was specified before the data were collected, and the analytic plan was pre-specified. This study was approved by the Institutional Review Board of the Institute of Tropical Medicine (IRB/RR/AC/134; 2 July 2021).

### 2.2. Description of Material

All data were collected using a structured online questionnaire (Appendix A), programmed in Microsoft Forms and available in Dutch, French and English, covering participants’ characteristics and food choice motives [22]. Self-reported questions on participant’s characteristics included socio-demographic information (age, sex, highest educational level attained, employment status, marital status, residential area and income level), anthropometric measures (self-reported body height and weight to calculate BMI) and information on COVID-19 (if they have/had been infected or someone from the household, self-isolated, and vaccinated). 

Food choice motives were evaluated using the Western Balkan Countries version of the Food Choice Questionnaire (FCQ) [23], originally developed by Steptoe et al. [24] in English and for the purpose of this study translated to Dutch and French. The original 36-items FCQ included 36 items and nine subscales related to food choice motives: health (6 items), mood (6 items), convenience (5 items), sensory appeal (4 items), natural content (3 items), price (3 items), weight control (3 items), familiarity (3 items), and ethical concerns (3 items). Each question followed the format “it is important to me that the food I eat on a typical day…”, and answers were scored from 1 to 5 indicating the following: 1 = strongly disagree; 2 = disagree; 3 = neither agree nor disagree; 4 = agree; and 5 = strongly agree. Instructions were given to assess food choice motives the year before (i.e., 2019) and during the years of the COVID-19 pandemic (i.e., 2020–2021), so that for each of the nine subscales, two scores were computed by averaging ratings for the individual items before and during the pandemic, respectively, and subsequently the difference of score for during and before. A difference above 0 indicated higher importance of that motive during the pandemic as compared to before, and vice versa, a difference below 0 represented the motive had lower importance during the pandemic. 

### 2.3. Statistical Analysis

General participants’ characteristics were presented as means and standard deviations for continuous variables and as counts and percentages for categorical variables. For descriptive purposes of the food choice motives, the mean scores of the nine subscales during the pandemic were compared with those from before with their mean difference tested using paired *t*-tests. 

To explore further the influence of participants’ characteristics on the changes in food choice motives, linear mixed regression analyses were performed for each motive observed to change using the score before and during (continuous) as the outcome and the participant’s characteristics (categorical) as explanatory variables, including a two time-point variable (i.e., with 0 for before and 1 for during) and a random intercept, and time-point interactions. Participants characteristics considered were those frequently reported to influence food choice motives, including socio-demographic background variables, such as age, sex and BMI [25], and food environment indicators [26], such as the residential area, and also the cultural context disruption by COVID-19 infection and associated compulsory isolation. Sociocultural factors such as education, employment, income, and marital status were not included due to low variability across the answer options, nor was COVID-19 vaccination status. Associations were reported as beta regression coefficients and their 95% confidence intervals (CIs), allowing us to identify which participant’s characteristics are the most important for a certain change in food choice motives. All statistical analyses were performed using R version 4.0.2, and the level of significance was set at *p* < 0.006 after applying Bonferroni correction for multiple comparisons.

## 3. Results

### 3.1. Participants

A total of 428 participants completed the online questionnaire (77.6% in Dutch, 5.6% in French, and 16.8% in English), and after excluding one participant younger than 18, information from 427 participants was analyzed. Participants’ characteristics for the total study population, also stratified by sex, are presented in Table 1. Most of the study population was female (73.5%), and the age ranged from 18 to 77 years (mean 41.2 years (SD 14.5)). When compared to women, male respondents were older, and reported more frequently to have completed at most secondary school, have retired, have a higher than average net salary, and were married/cohabited. Men also had a higher BMI and higher proportions of overweight and obesity. In turn, female respondents reported working in healthcare-related jobs more frequently than men, as well as living in urban areas. No sex differences were observed in the information reported regarding COVID-19 history.

### 3.2. Food Choice Motives before and during the COVID-19 Pandemic

Table 2 presents the importance of the food choice motives before and during the pandemic for the total sample and stratified by sex. Overall, the most important motive for food choice was sensory appeal both before and during the COVID-19 pandemic, albeit the score for this motive was slightly, but non-significantly, lower during the pandemic (mean score of 3.98 (SD 0.48) during vs. 4.02 (SD 0.51) before). Convenience and health also ranked high among motives for food choice, but only changes in the health motive were observed, being significantly more important during the pandemic, although modest changes in mean scores (mean score of 3.69 (SD 0.60) during vs. 3.64 (SD 0.59) before), and in particular 37.0% declared that health was more important while facing the pandemic. Mean scores of mood (3.41 (SD 0.71) vs. 3.32 (SD 0.58)), natural content (3.35 (SD 0.84) vs. 3.26 (SD 0.85)) and weight control (3.33 (SD 0.79) vs. 3.25 (SD 0.76)) were significantly higher during the pandemic as compared to before, with 43.3% of the participants declaring that mood was more important, 34.7% natural content and 36.5% weight control. The latter, weight control, was the motive with the greatest mean change for men, with 41.4% of the male sample reporting an increase in this motive. In women, the mean scores of the food choice motives mood (3.44 (SD 0.70) vs. 3.33 (SD 0.56)) and natural content (3.37 (SD 0.82) vs. 3.27 (SD 0.82)) were reported to be significantly more important during the pandemic as compared to before, with 44.3% of the women reporting higher importance of mood in their food choices and 35.0% natural content. Although no significant mean differences in scores, health was reported to be more important in making food choices for 37.6% of the women and weight control for 34.7%.

### 3.3. Socio-Demographic Determinants of Food Choice Motives and Their Changes Induced by the Pandemic

Table 3 presents the exploratory findings regarding the associations of participants’ characteristics with the food choices motives and their changes during the pandemic as compared to before.

As observed in Table 2, mood, followed by natural content, weight control, and health were the motives that modestly changed during the pandemic. However, when accounting for socio-demographic variables in a multivariate model, only the motive related to the natural content remained significantly influenced by the pandemic (Beta 0.34; 95%CI: 0.14, 0.54).

When compared to men, women scored higher in most food choice motives, and particularly health (0.29; 0.16, 0.41) and sensory appeal (0.21; 0.10, 0.31), both before and during the pandemic, while familiarity was significantly more important to men (−0.34; −0.50, −0.17). With increasing age, natural content (0.19; 0.13, 0.25) and ethical concerns (0.19; 0.12, 0.25) were more important both before and during the pandemic, while price (−0.09; −0.14, −0.04) and familiarity (−0.08; −0.14, −0.03) were more important at younger ages. These age-related differences in food choice motives were significantly reduced during the pandemic for natural content (−0.06; −0.10, −0.03). When compared to participants from semirural areas, respondents from urban areas scored higher in the motive related to ethical concerns (0.26; 0.10, 0.42), and during the pandemic, the motive related to health (0.10; 0.03, 0.16) became more important in their food choices as compared to before, which was not the case for those from the semirural areas. Natural content, although not statistically significant (0.15; −0.01, 0.31), was also more important to participants from the urban area both before and during the pandemic, while familiarity was scored lower (−0.13; −0.28, 0.02). Mood (0.19; 0.06, 0.32) was significantly more important to participants with overweight, and this was for both before and during the pandemic.

Comparison by infection status and the need for self-isolation revealed no significant differences either before or during the pandemic, with the exception of sensory appeal (0.21; 0.06, 0.35), which for those who had been infected for both before and during the pandemic, despite an overall decreasing level of importance during the pandemic.

## 4. Discussion

The same FCQ has been distributed in Croatia before [22], hence its distribution in Belgium serves as a pilot project for future larger international studies on food choice motives in the COVID-19 pandemic and during post-COVID-19 times. This also allows for a direct descriptive comparison of the results, showing that in both Croatia and Belgium, the majority of the food choice motives, with the exception of sensory appeal and mood (Croatia only) and familiarity (Belgium only), were mostly more important during the pandemic as compared to before, with the largest mean difference for weight control in Croatia and for mood and natural content in Belgium. Food choice motives were different across population subgroups with the main differences being determined by sex, with women scoring higher in the majority of motives except for the motive of familiarity, as inquired by our study. Additionally, differences in motives were driven by age with more importance given to natural content and ethical concerns and less to price and familiarity as the age of the respondent increases, by residential area with more importance to ethical concerns for residents in urban areas. Furthermore, the COVID-19 pandemic influence on the importance of food choice motives was observed to be lower with increasing age (particularly for natural content), but higher for the participants living in the urban area (particularly for health).

In our sample of Belgian adults, the level of importance increased the most, though only modestly, during the pandemic for mood and natural content, and particularly in women. This observation about the importance of natural content agrees with previous reports from Polish female adolescents [19], Croatian women [22], and overall adults from France [20] and UK [21], and that of mood with reports from France [20] and UK [21] only. A further comparison of the findings with those of other studies [19,20,21,22] confirms that food choice motives of health and weight control were more important during the pandemic as compared to before. This is contrary to previous reports, where weight gain during first months of confinement was often reported in adults, in particular among individuals with overweight or obesity and who indicated having increased their snacking and reduced physical activity [27], and this was also reported for Belgium [16]. It remains questionable whether this disrupted weight management might persist throughout and post pandemic, because of the higher importance of health and weight control as food choice motives during the pandemic for more than one-third of our sample. According to our and also earlier observations on food choice motives, we can at best infer that this modest increase in the importance of health and weight control as a food choice motive may suggest a growing awareness of the importance of healthy food choices during the pandemic, at least in some participants.

In general, among the consumers reporting changes in food-related behaviors, a majority claimed to be more concerned about the healthiness along with the sustainability of their food products, an existing tendency of the recent years [28]. This tendency seems to be accelerated during the COVID-19, and in particular, paying attention to body weight, and choosing more unpacked or recyclable/biodegradable packed foods and local production [29], of which this latter was highly promoted in Belgium during the COVID-19 pandemic. In contrast, our study, though not fully capturing all dimensions of sustainability [30], found half of the respondents declaring no change in the food choice motive of ethical concerns, i.e., the motive closest to environmental sustainability covering food packaging, country of origin and politically approved countries, and showing, in general, a higher level of importance with increasing age and those living in the urban area. In contrast to earlier findings [31,32,33]; however, no evidence was observed for a differential impact of the pandemic on healthy and sustainable food choices by sex and overweight status, whereas according to our data, the extent and/or direction of a change in the level of importance mainly differed by age (natural content) and residential area (health). Still, our results suggest that for a moderate share of the population, food choice motives have a habitual and enduring character, since for each food choice motive between a quarter and half of the respondents declared no change during the pandemic as compared to before, which is in line with previous results on food-related behaviors during the pandemic conducted in different contexts [29,31,32].

Sensory appeal, including taste, smell, texture, and appearance of foods, plays an important functional role in defining our food likes and dislikes, and hereby is a major determinant of our food choice and intake, i.e., selecting and consuming more of foods that are liked and rarely the disliked foods [34]. This acquisition of food preferences is highly relying on mere exposure to a variety of different tastes, smells, textures and appearances, and learning to accept them. In addition, the level of importance for sensory appeal as a major food choice motive seemed to be unaffected by a pandemic context with disrupted food access (closures of bars, restaurants), increased food prices and loss of disposable household income, as suggested by our study. This may be explained by flavor-nutrient associations, i.e., the association between sensory characteristics of a food and its post-ingestive effects, which mainly guide our food choices [35]. Nevertheless, a growing body of evidence has recognized the importance of these flavor-nutrient associations to also steer appetite and calorie selection, highlighting the importance of maintain sensory quality in healthy foods [34,35].

Our findings complement earlier observations from the Corona Cooking Survey, as conducted in a total of 38 countries worldwide, including Belgium, during the earlier phases of the pandemic (April–June 2020), which suggested that stay-at-home policies, as induced by the COVID-19 pandemic, and associated perceptions of having more time were most clearly associated with increases in the willingness to plan, select and prepare healthier foods among women and men [36]. This in particular matches the modestly increased level of importance for the food choice motives of natural content, weight control and health, as observed in our study. Moreover, the second edition of the Belgian Corona Cooking Survey, conducted in November–December 2020, showed that about one-third of the respondents more often think about what they eat and choose healthy foods while cooking [37]. Additionally, since the pandemic, most respondents of the Corona Cooking Survey do experience cooking as less time consuming and less stressful, and more than one-third find cooking more enjoyable associated with higher self-ratings of cooking skills for almost a quarter, particularly for preparing cakes/cookies (18.1%) and bread (13.7%) [38]. It is possible, therefore, that more enjoyment in cooking might partly explain the observed positive influence of the pandemic on the food choice motive of mood corresponding to selecting foods that help to cheer up and cope with stress.

The findings of this study are based on self-reports of self-perceived food choice motives before and during the COVID-19 pandemic; hence, they are vulnerable to an inevitable recall bias and the biases of overreporting socially desirable behaviors and underreporting those undesirable ones. Additionally, at the personal level, we cannot exclude the occurrences that are not directly related to the COVID-19 pandemic, but could have influenced food choice motive changes, nor the differential implications on daily routines (for example, switching to home office or temporary unemployment). Data were collected during the summer and early fall of 2021 when Belgium was in the period of the third wave and many of the participants were taking up the vaccination program but were also tired of COVID-19 and its associated implications on daily routines. This may have influenced response rates. Furthermore, our recruitment strategy, relying on our social media channels and professional networks, introduced a selection bias towards academic profiles that can be observed in the elevated proportion of higher education group, and the unequal response between Belgian regions being Flanders over-represented. However, with this underrepresentation of the low-educated and low-income strata, caution must be applied, as the pandemic’s influence on the food choice motive price could not have been detected, which is contrary to a previous study that found about 10% of the Belgians often or sometimes feared food insecurity during confinement [39].

Nevertheless, our data provide complementary insights into the impact of the COVID-19 pandemic on food choice motives at a time when people were COVID fatigued. To the best of our knowledge, this is the first study demonstrating no time difference in the food choice motives by infection status and by self-isolation status.

## 5. Conclusions

Sensory appeal, convenience and health were ranked as the top three food choice motives before and during the COVID-19 pandemic, among a sample of Belgian adults (mainly female, Flanders’ residents, and highly educated). In particular, the level of importance was significantly higher for mood and natural content, followed by weight control and health during the pandemic as compared to before, although only modest changes. However, changes in the level of importance were observed in both directions, while also a moderate share of the respondents declared no change, suggesting some persistence of food choice motives. Further activities within public health monitoring should be undertaken to fully understand the implications of COVID-19 on food choice motives together with people’s food behaviors and consumption. At the same time, actions from policy makers are needed to further tailor health promotion programs for vulnerable population subgroups aimed at preventing changes towards unhealthier diets driven by a severe disruption in individual’s daily routines and to further support sustainable food systems.

## Figures and Tables

**Table 1 foods-11-00842-t001:** General participants’ characteristics, for the total sample and according to sex ^1^.

	Overall (*n* = 427)	Males (*n* = 111, 26%)	Females (*n* = 314, 73.5%)
Socio-demographic characteristics			
Age (years), mean (SD)	41.2 (14.5)	44.5 (15.1)	40.0 (14.2)
Highest educational qualification, *n* (%)			
≤Secondary school	79 (18.5)	32 (28.8)	47 (15.0)
Bachelor’s degree	152 (35.6)	30 (27.0)	121 (38.5)
Master’s degree	161 (37.7)	40 (36.0)	120 (38.2)
Doctorate degree	35 (8.2)	9 (8.1)	26 (8.3)
Employment status, *n* (%)			
Employed, full/part-time	327 (76.6)	84 (75.7)	242 (77.1)
Student	50 (11.7)	10 (9.0)	39 (12.4)
Retired	36 (8.4)	14 (12.6)	22 (7.0)
Unemployed	14 (3.3)	3 (2.7)	11 (3.5)
Working in or being closely related to the healthcare system, *n(%)*	161 (37.7)	24 (21.6)	136 (43.3)
Missing	75 (17.6)	18 (16.2)	56 (17.8)
Monthly net income, *n* (%) ^2^			
≤minimum	72 (16.9)	14 (12.6)	57 (18.2)
Minimum-average	288 (67.4)	74 (66.7)	213 (67.8)
≥average	41 (9.6)	19 (17.1)	22 (7.0)
No independent income	26 (6.1)	4 (3.6)	22 (7.0)
Marital status, *n* (%)			
Unmarried	151 (35.4)	36 (32.4)	115 (36.6)
Married/cohabitation	246 (57.6)	70 (63.1)	175 (55.7)
Divorced/separated	24 (5.6)	5 (4.5)	18 (5.7)
Widowed	6 (1.4)	0 (0)	6 (1.9)
Residential area, *n* (%)			
Urban area (city)	258 (60.4)	61 (55.0)	195 (62.1)
Semirural area (countryside)	169 (39.6)	50 (45.0)	119 (37.9)
Anthropometrics			
Body weight (kg), mean (SD)	70.6 (14.4)	80.5 (15.2)	67.0 (12.3)
Missing, *n (%)*	9 (2.1)	1 (0.9)	8 (2.5)
Height (cm), mean (SD)	171 (8.24)	179 (6.45)	168 (6.65)
BMI (kg/m^2^), mean (SD)	24.1 (4.13)	25.1 (4.39)	23.8 (3.99)
Missing, *n (%)*	9 (2.1)	1 (0.9)	8 (2.5)
Underweight (<8.5), *n* (%)	11 (2.6)	0 (0)	11 (3.5)
Normal weight (18.5–24.9), *n* (%)	264 (61.8)	61 (55.0)	201 (64.0)
Overweight (25–29.9), *n* (%)	106 (24.8)	35 (31.5)	71 (22.6)
Obese (≥30), *n* (%)	37 (8.7)	14 (12.6)	23 (7.3)
Information on COVID-19			
COVID-19 infection, *n* (%)	56 (13.1)	16 (14.4)	40 (12.7)
Method used for virus infection confirmation in those who tested positive, *n* (%)			
By PCR test	39 (9.1)	9 (8.1)	30 (9.6)
By rapid antigen test	1 (0.2)	1 (0.9)	0 (0)
By serology test	3 (0.7)	2 (1.8)	1 (0.3)
Not confirmed by any of the above-mentioned methods	13 (3.0)	4 (3.6)	9 (2.9)
Confirmed COVID-19 infection in household members, *n* (%)	66 (15.5)	21 (18.9)	45 (14.3)
Self-isolation due to COVID-19 preventive measures, *n* (%)	191 (44.7)	53 (47.7)	137 (43.6)
COVID-19 vaccination, *n* (%)			
Yes, fully vaccinated	331 (77.5)	85 (76.6)	244 (77.7)
Partly	78 (18.3)	23 (20.7)	55 (17.5)
No	18 (4.2)	3 (2.7)	15 (4.8)

^1^ 2 (0.5%) participants reported sex as X; ^2^ minimum net salary was set to 1500 euros and average net salary to 3550 euros. Source: Author’s elaboration of data from the online-based survey.

**Table 2 foods-11-00842-t002:** Food choice motives before and during the COVID-19 pandemic for the study population, and stratified by sex (*n* = 427) ^1,2^.

	Before Mean (SD)	During Mean (SD)	Difference Mean (SD)	Pvalue ^3^	Increased *n* (%)	Unchanged *n* (%)	Decreased *n* (%)
Total (*n* = 427)							
Sensory appeal	4.02 (0.51)	3.98 (0.48)	−0.03 (0.34)	0.039	115 (26.9%)	155 (36.3%)	157 (36.8%)
Convenience	3.65 (0.61)	3.66 (0.69)	0.01 (0.49)	0.540	153 (35.8%)	136 (31.9%)	138 (32.3%)
Health	3.64 (0.59)	3.69 (0.60)	0.05 (0.33)	0.003 *	158 (37.0%)	142 (33.3%)	127 (29.7%)
Price	3.44 (0.67)	3.46 (0.73)	0.02 (0.44)	0.380	140 (32.8%)	169 (39.6%)	118 (27.6%)
Mood	3.32 (0.58)	3.41 (0.71)	0.10 (0.43)	<0.001 *	185 (43.3%)	110 (25.8%)	132 (30.9%)
Natural content	3.26 (0.85)	3.35 (0.84)	0.09 (0.47)	<0.001 *	148 (34.7%)	181 (42.4%)	98 (23.0%)
Weight control	3.25 (0.76)	3.33 (0.79)	0.08 (0.50)	<0.001 *	156 (36.5%)	180 (42.2%)	91 (21.3%)
Ethical concerns	3.18 (0.85)	3.22 (0.87)	0.04 (0.44)	0.041	116 (27.2%)	219 (51.3%)	92 (21.5%)
Familiarity	2.91 (0.74)	2.91 (0.81)	0.00 (0.52)	0.876	129 (30.2%)	165 (38.6%)	133 (31.1%)
Men (*n* = 111)							
Sensory appeal	3.86 (0.50)	3.83 (0.46)	−0.04 (0.33)	0.250	24 (21.6%)	48 (43.2%)	39 (35.1%)
Convenience	3.53 (0.63)	3.57 (0.71)	0.04 (0.48)	0.406	40 (36.0%)	37 (33.3%)	34 (30.6%)
Health	3.44 (0.61)	3.48 (0.62)	0.04 (0.31)	0.174	39 (35.1%)	38 (34.2%)	34 (30.6%)
Price	3.35 (0.62)	3.43 (0.67)	0.07 (0.46)	0.101	41 (36.9%)	43 (38.7%)	27 (24.3%)
Mood	3.29 (0.63)	3.35 (0.73)	0.06 (0.41)	0.149	45 (40.5%)	25 (22.5%)	41 (36.9%)
Natural content	3.21 (0.94)	3.30 (0.91)	0.08 (0.49)	0.073	38 (34.2%)	49 (44.1%)	24 (21.6%)
Weight control	3.13 (0.83)	3.25 (0.84)	0.12 (0.54)	0.024	46 (41.4%)	42 (37.8%)	23 (20.7%)
Ethical concerns	3.08 (0.88)	3.14 (0.96)	0.05 (0.47)	0.224	36 (32.4%)	47 (42.3%)	28 (25.2%)
Familiarity	3.16 (0.70)	3.08 (0.77)	−0.07 (0.46)	0.103	25 (22.5%)	47 (42.3%)	39 (35.1%)
Women (*n* = 314)							
Sensory appeal	4.07 (0.50)	4.04 (0.48)	−0.03 (0.35)	0.103	91 (29.0%)	107 (34.1%)	116 (36.9%)
Convenience	3.69 (0.60)	3.69 (0.69)	0.00 (0.49)	0.909	111 (35.4%)	99 (31.5%)	104 (33.1%)
Health	3.71 (0.56)	3.76 (0.57)	0.05 (0.33)	0.009	118 (37.6%)	103 (32.8%)	93 (29.6%)
Price	3.48 (0.68)	3.48 (0.75)	0.00 (0.43)	0.966	99 (31.5%)	125 (39.8%)	90 (28.7%)
Mood	3.33 (0.56)	3.44 (0.70)	0.11 (0.44)	<0.001 *	139 (44.3%)	85 (27.1%)	90 (28.7%)
Natural content	3.27 (0.82)	3.37 (0.82)	0.09 (0.47)	<0.001 *	110 (35.0%)	131 (41.7%)	73 (23.2%)
Weight control	3.30 (0.73)	3.36 (0.76)	0.07 (0.49)	0.014	109 (34.7%)	137 (43.6%)	68 (21.7%)
Ethical concerns	3.21 (0.84)	3.25 (0.84)	0.04 (0.43)	0.083	80 (25.5%)	171 (54.5%)	63 (20.1%)
Familiarity	2.83 (0.72)	2.85 (0.82)	0.01 (0.53)	0.618	103 (32.8%)	117 (37.3%)	94 (29.9%)

^1^ Cronbach’s alpha, a measure of internal consistency, for the food choice motives before: Health 0.81; Mood 0.72; Convenience 0.75; Sensory appeal 0.66; Natural content 0.86; Price 0.72; Weight control 0.78; Familiarity 0.64; Ethical concerns 0.75, during: Health 0.83; Mood 0.85; Convenience 0.83; Sensory appeal 0.70; Natural content 0.88; Price 0.76; Weight control 0.82; Familiarity 0.74; Ethical concerns 0.81. ^2^ Score range from 1 to 5. ^3^ Paired *t*-test using a Bonferroni corrected alpha level of 0.006 with an asterisks (*) to indicate the statistical significant differences. Source: Author’s elaboration of data from the online-based survey.

**Table 3 foods-11-00842-t003:** Exploratory linear mixed model analysis of participants’ characteristics and food choice motives ^1^.

	Sensory Appeal	Convenience	Health	Price	Mood	Natural Content	Weight Control	Ethical Concerns	Familiarity
Time, before vs. during	−0.08	0.04	0.05	0.18	0.17	0.34 *	0.20	0.09	0.12
	(−0.23; 0.06)	(−0.17; 0.26)	(−0.09; 0.19)	(−0.01; 0.36)	(−0.02; 0.35)	(0.14; 0.54)	(−0.01; 0.42)	(−0.10; 0.28)	(−0.10; 0.34)
Sex, females vs. males	0.21 *	0.15	0.29 *	0.10	0.03	0.11	0.20	0.19	−0.34 *
	(0.10; 0.31)	(0.00; 0.29)	(0.16; 0.41)	(−0.05; 0.26)	(−0.11; 0.17)	(−0.07; 0.29)	(0.03; 0.38)	(0.00; 0.37)	(−0.50; −0.17)
Age, per 10 years	−0.03	−0.05	0.06	−0.09 *	−0.06	0.19 *	0.08	0.19 *	−0.08 *
	(−0.07; 0.00)	(−0.10; −0.01)	(0.02; 0.10)	(−0.14; −0.04)	(−0.10; −0.01)	(0.13; 0.25)	(0.02; 0.13)	(0.12; 0.25)	(−0.14; −0.03)
Residential area, urban vs. semirural	0.00	−0.03	−0.01	−0.03	0.05	0.15	−0.01	0.26 *	−0.13
	(−0.10; 0.09)	(−0.16; 0.09)	(−0.13; 0.10)	(−0.16; 0.11)	(−0.07; 0.18)	(−0.01; 0.31)	(−0.17; 0.14)	(0.10; 0.42)	(−0.28; 0.02)
BMI status, BMI ≥ 25 vs. < 25	0.13	0.04	−0.13	0.17	0.19 *	−0.11	0.00	−0.05	0.12
	(0.03; 0.23)	(−0.10; 0.17)	(−0.25; −0.01)	(0.03; 0.31)	(0.06; 0.32)	(−0.28; 0.06)	(−0.16; 0.16)	(−0.22; 0.12)	(−0.03; 0.28)
COVID−19 infection, yes vs. no	0.21 *	0.13	0.14	0.14	0.05	−0.01	0.16	−0.06	−0.01
	(0.06; 0.35)	(−0.06; 0.33)	(−0.03; 0.32)	(−0.06; 0.35)	(−0.15; 0.24)	(−0.26; 0.24)	(−0.07; 0.40)	(−0.31; 0.19)	(−0.24; 0.22)
Self-isolation, yes vs. no	−0.06	−0.02	−0.01	0.06	0.11	0.11	0.01	0.05	0.09
	(−0.17; 0.04)	(−0.16; 0.12)	(−0.14; 0.11)	(−0.09; 0.21)	(−0.03; 0.25)	(−0.07; 0.29)	(−0.16; 0.18)	(−0.13; 0.23)	(−0.08; 0.25)
Time ∗ Sex	0.01	−0.03	0.00	−0.09	0.03	−0.01	−0.06	−0.04	0.08
	(−0.06; 0.09)	(−0.14; 0.08)	(−0.08; 0.07)	(−0.18; 0.01)	(−0.07; 0.13)	(−0.11; 0.09)	(−0.17; 0.05)	(−0.13; 0.06)	(−0.04; 0.19)
Time ∗ Age	0.01	−0.01	−0.01	−0.03	−0.03	−0.06 *	−0.03	−0.03	−0.04
	(−0.01; 0.04)	(−0.04; 0.03)	(−0.04; 0.01)	(−0.06; 0.00)	(−0.06; 0.00)	(−0.10; −0.03)	(−0.06; 0.01)	(−0.06; 0.00)	(−0.08; 0.00)
Time ∗ Residential area	−0.02	0.00	0.10 *	0.00	0.08	0.04	0.05	0.10	−0.07
	(−0.09; 0.05)	(−0.10; 0.09)	(0.03; 0.16)	(−0.09; 0.09)	(0.00; 0.17)	(−0.05; 0.14)	(−0.04; 0.15)	(0.01; 0.18)	(−0.17; 0.03)
Time ∗ BMI status	−0.01	0.08	0.02	−0.01	−0.04	0.02	0.08	0.04	0.05
	(−0.08; 0.06)	(−0.02; 0.18)	(−0.05; 0.09)	(−0.10; 0.08)	(−0.14; 0.05)	(−0.08; 0.11)	(−0.02; 0.19)	(−0.05; 0.13)	(−0.06; 0.15)
Time ∗ COVID-19 infection	−0.04	−0.13	−0.08	0.02	−0.03	0.08	−0.05	−0.05	0.05
	(−0.15; 0.06)	(−0.28; 0.02)	(−0.18; 0.02)	(−0.11; 0.16)	(−0.16; 0.10)	(−0.07; 0.22)	(−0.20; 0.11)	(−0.19; 0.08)	(−0.10; 0.21)
Time ∗ Self-isolation	0.02	0.04	0.01	0.04	0.02	−0.06	−0.04	0.06	−0.02
	(−0.06; 0.09)	(−0.07; 0.15)	(−0.07; 0.08)	(−0.06; 0.14)	(−0.08; 0.11)	(−0.16; 0.04)	(−0.15; 0.07)	(−0.03; 0.16)	(−0.13; 0.09)

^1^ beta regression coefficients and corresponding 95% confidence intervals of the multi-variate adjusted model in 416 respondents with complete reliable data. * *p*-value ≤ 0.006. Source: Author’s elaboration of data from the online-based survey.

## Data Availability

The data generated or analyzed during this study are available from the corresponding author on reasonable request.

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
