# Peer review of "Food Choice Motives and COVID-19 in Belgium"

_foods, 2022, doi:10.3390/foods11060842_

Round 1

Reviewer 1 Report

The manuscript brings very interesting results, however it has some flaws

Below some considerations:

Abstract:

Line 17: What period was before?

This issue is not clear in the whole manuscript.

Line 23:  “Convenience and health also ranked among the main motives” … in what period?

Introduction

Line 42: “Yet, the opposite has been also observed, for instance, a decrease in the frequency of ordered food and the fast-food consumption [3],” ... Are there other works that bring this statement?

Results

Line 145: “No differences were observed in the information reported regarding COVID-19 history.”....this sentence is not clear

Table 2

Put in table 2 the significant level.

Cronbach’s alpha before???? And during? What it means?

Discussion

Sensory appeal emerged as an important food choice motive. In a pandemic context this result needs a discussion. Sensory appeal highlights the importance to mantain sensory quality in healthy foods.

Reviewer 2 Report

The article deals with important research issues on the food choice motives and COVID-19 in Belgium. However, I suggest some comments to improve the structure of the manuscript.

Abstract:

This empirical study will analyse a sample of 427 eligible respondents from Belgium: Please, specify the area in Belgium.

In the Abstract section the main recommendations are for policy makers.

1. Introduction

Lines 40-41: Mandatory isolation and remote work severely disrupt the individuals’ 40 daily routines, likely resulting in decreasing levels of physical activity and increasing sedentary behavior [1].

Please, report recent study

Lines 47-48: Early findings from Belgium. Please give a brief description of these results

Lines 68-73: The aim of the present study is… I suggest to report the Research Questions (RQs) as following:

RQ1:…..

RQ2:….

RQ3:….

2. Theoretical framework

I suggest adding this section to report the results of important studies recently conducted on the subject:

Haleem, A.; Javaid, M.; Vaishya, R. Effects of COVID-19 pandemic in daily life. Curr. Med. Res. Pract. 2020, 10, 78-79;

Górnicka, M.; Drywie ´n, M.E.; Zielinska, M.A.; HamuÅ‚ka, J. Dietary and Lifestyle Changes During COVID-19 and the Subsequent Lockdowns among Polish Adults: A Cross-Sectional Online Survey PLifeCOVID-19 Study. Nutrients 2020, 12, 2324;

Martinez-Ferran, M.; de la Guía-Galipienso, F.; Sanchis-Gomar, F.; Pareja-Galeano, H. Metabolic Impacts of Confinement during the COVID-19 Pandemic Due to Modified Diet and Physical Activity Habits. Nutrients 2020, 12, 1549;

Mattioli, A.V.; Sciomer, S.; Cocchi, C. Quarantine during COVID-19 outbreak: Changes in diet and physical activity increase the risk of cardiovascular disease. Nutr. Metab. Cardiovasc. Dis. 2020, 9, 1409–1417;

Durante, K.M.; Laran, J. The effect of stress on consumer saving and spending. J. Mark. Res. 2016, 53, 814–828;

Muscogiuri, G.; Barrea, L.; Savastano, S.; Colao, A. Nutritional recommendations for CoVID-19 quarantine. Eur. J. Clin. Nutr. 2020, 74, 850–85

2. 3Materials and methods

2.1. 3.1 Study design and study population

Lines 76-77: How was the online survey was launched? Please specify in a better way

2.2. 3.2 Description of material

Lines 88-96: Describe in detail the content and structure of the instrument used to collect information.

2.3. 3.3 Statistical analysis and not analyses

Lines 113-133: The descriptive statistics and the linear mixed regression analysis have find large application in the studies related to consumer choices, please, see the works suggested for the Theoretical framework to explain in a better way the influence of participants’ characteristics on the changes in food 118 choice motives.

3. 4. Results

Add theoretical aspects in a new paragraph (2. Theoretical framework) that allow the results to be discussed with other previous investigations.

4. 5. Discussion

The literature review is too modest to work out a research gap and to discuss the results of your study in relation to findings of previous studies in the field.

5. 6. Conclusions and Political Implications

I suggest to extend this paragraph that should include recommendations for further research, for policy makers, but also the study’s limitations.

Tables

Finally, the quality of Tables 2 and 3 must be improved

Please, enter a Table 2 and 3 with asterisk to highlight significance of Food Choice Motives before and during the COVID-19 pandemic.

Please report the Source at at the foot of the tables

Authors’ elaboration of data from online survey

Reviewer 3 Report

Dear authors, I read with great interest your manuscript, it was well-written, has good English translation, strong statistics and design, original aim, proper questionnaire and type of research, interesting results and conclusions, also recent cited references.

It is relevant and interesting. Our lifestyle and especially diet, cam influence our health status and the motives for choosing some food products can be evaluated and than educated in order to have a proper diet.

The originality comes from the links between Covid-19 impact and KAP survey upon the consumers attitudes towards food. Also the questionnaires used are recent, valid and complex.

Compared with other published material it add to the subject area links between Covid-19, life style, psycho-social risk factors, consumers attitudes in stress situations, evaluation of a risk profile for consumers etc.

The paper is well written, the text was clear and easy to be read, the manuscript was well-written, proper English as well, concrete infos.

The conclusions were consistent with the evidence and with good arguments, proper discussion chapter, recent references cited.

They address the main question, in detail and based on strong statistics and good evidence, even though was an observational study it can be take into account, for a pilot study on this subject.

Reviewer 4 Report

Dear Authors,

The manuscript (foods-1610266) presented for review is very interesting. I recommend it for minor corrections.

Authors, Please note and address the following comments:

This manuscript differs from other similar manuscripts of its kind created during the pandemic by questions for COVID-19-related information. It would be very interesting if the authors added a questionnaire as supplementary material. It is also a pity that these data (related consumers to Covid) were not analyzed with the obtained results. Perhaps the authors are planning another article, and in this, they limited themselves only to the motives of the food choice.

Introduction: The background of this study is poor. There is a lot of research related to human nutrition during a pandemic, and this also applies to Europe. The authors should show why this study is important and whether it fills any gaps in knowledge about this topic. Authors can find many articles on this topic in "Nutrients", "Foods" and other journals.

Keywords: In my opinion, keywords should be rethought.

Limitation: I propose to separate the limitations and strengths of these results.  

Discussion: The authors discuss their results with other studies, but do not try to explain why consumers in Belgium did not change their eating habits despite the health risks during the pandemic.

References: References are not cited according to journal rules. Publications from MDPI provide information on how to properly cite. Authors may also find this information in the authors' guide.

I would suggest adding more studies and surveys from 2020 and 2021,

I believe it addresses an important area of research in an international context.

Reviewer
